# Solution-Processable Growth and Characterization of Dandelion-like ZnO:B Microflower Structures

**Selma Erat** [1,2,3,*], **Artur Braun** [4], **Samed Çetinkaya** [1,2,3,*], **Saadet Yildirimcan** [1,2,3], **Ahmet Emre Kasapoğlu** [5], **Emre Gür** [6], **Ersan Harputlu** [7] **and Kasım Ocakoglu** [7]

[1] Department of Medical Services and Techniques, Vocational School of Technical Sciences, Mersin University, Mersin 33340, Turkey; saadetyildirimcan@mersin.edu.tr
[2] Advanced Technology, Research and Application Center, Mersin University, Mersin 33340, Turkey
[3] Department of Nanotechnology and Advanced Materials, Institute of Science, Mersin University, Mersin 33340, Turkey
[4] Laboratory for High Performance Ceramics, Empa—Swiss Federal Laboratories for Materials Science and Technology, CH-8600 Dubendorf, Switzerland; artur.braun@empa.ch
[5] East Anatolia High Technology Application and Research Center, Atatürk University, Erzurum 25240, Turkey; aemre.kasapoglu@atauni.edu.tr
[6] Department of Physics, Faculty of Science, Atatürk University, Erzurum 25240, Turkey; emregur@atauni.edu.tr
[7] Department of Engineering Basic Sciences, Faculty of Engineering, Tarsus University, Mersin 33400, Turkey; ersanharputlu@tarsus.edu.tr (E.H.); kasim.ocakoglu@tarsus.edu.tr (K.O.)
[*] Correspondence: selma.erat@mersin.edu.tr (S.E.); samedcetinkaya@mersin.edu.tr (S.Ç.); Tel.: +90-324-361-00-01 (S.E. & S.Ç.); Fax: +90-324-361-01-53 (S.E. & S.Ç.)

**Abstract:** Intrinsic and dandelion-like microflower nano-rod structures of boron-doped ZnO thin films were synthesized with an ecofriendly and cost-effective chemical bath deposition technique from an aqueous solution of zinc nitrate hexahdyrate [$Zn(NO_3)_2 \cdot 6H_2O$] as a precursor solution and boric acid as a doping solution. The boron concentrations were 0.1, 0.3, 0.5, 1.0, 3.0, 5.0, and 7.0 by volume. Scanning electron micrographs showed that doping with boron appears to hinder the vertical alignment of crystallites. Additionally, independent hexagonal nano-rod structures were observed to coalesce together to form dandelion-like structures on the film's surface. The atomic ratio of the elements was determined via the X-ray photoemission spectrum technique. There were no substantial changes in the vibration structure of the film upon doping in terms of the Raman spectra. The optical band gap of ZnO (3.28 eV) decreased with B doping. The band gap of the ZnO:B film varied between 3.18 and 3.22 eV. The activation energy of the ZnO was calculated as 0.051 eV, whereas that of the ZnO:B film containing 1.0% B was calculated as 0.013 eV at low temperatures (273–348 K), versus 0.072 eV and 0.183 eV at high temperatures (348–523 K), respectively. Consequently, it can be interpreted that the 1% B-doped ZnO, which has the lowest activation energy at both low and high temperatures, may find some application areas such as in sensors for gases and in solar cells.

**Keywords:** ZnO; dandelion; boron; micro rods

## 1. Introduction

Transparent conductive oxide (TCO) thin films, especially the compound semiconductor forms of the Group II–VI elements (such as ZnO and CdO, but also non-oxide films such as ZnS, CdS, ZnSe, CdSe, ZnTe, and CdTe) have been an interesting research topic due to their simple and cost-effective production processes [1]. Irrespective of their fabrication process, these semiconductors have n-type conductive behavior [1,2].

Among the TCOs, zinc oxide (ZnO) is popular because of its interesting structural, chemical, optical, and electrical properties [1–3]. It was one of the first studied wide band gap semiconductors with respect to its surface properties, optical properties, and gas sensing properties [1–4].

ZnO has potential for application in optoelectronics and spintronic devices due to its unique properties such as a wide band gap (3.37 eV) at room temperature and its high exciton binding energy (60 meV) [5–17]. Its practical application areas include light-emitting diodes, ultraviolet photodetectors, solar cells and collectors, electrochromic devices, gas sensors, hydrogen production and surface acoustic wave devices [1,9,12,14–16,18–20].

Researchers have intensively studied ZnO thin films and powders, including those doped with Cr, Mn, Fe, Co, Ni, Cu, Al, Ga, Cd, In, B, and N. The purpose of the dopant is to improve the structural, optical, and electrical properties of ZnO, and to adjust the band gap and super-cage-nano heterostructures with less cage mismatch [7,8,10–17,21–26]. Devices based on ZnO nanostructures are alternatives to Group III–V semiconductor compounds such as GaN-, and GaAs-based devices. Among the ZnO nanostructures, one dimensional nano-rods (1D) have been in the focus of current research in physics, chemistry, and material science due to their technological applications as well as their fundamental research importance. The growth of 1D structures such as nano-rods have been reported for ZnO doped with B, Al, Li, Mg, and In, for example, in [1,27]. The microscopic scenario is that Zn atoms in ZnO take their place in the crystal lattice. The electronic conductivity increases by providing extra electrons if they are added from Group III elements with higher valence and smaller ionic radii compared with the Zn cation [28]. The use of such one-dimensional (1D) nanostructures has been one of the alternative ways of producing hydrogen in photo-electrochemical cells, since it reduces electron recombination. Such nanostructures have advantages due to the smaller surface area and also have disadvantages due to a small contact surface [14,29].

There is a wide variety of fabrication methods for ZnO thin films such as molecular beam epitaxy (MBE), pulsed laser deposition (PLD), atomic layer deposition (ALD), radio frequency (RF) magnetron sputtering, chemical bath deposition (CBD), hydrothermal synthesis, sol-gel spin coating, sol-gel dip coating, and microwave synthesis [14–16,30–40]. Among all of these, chemical bath deposition (CBD) is a solution-based method that is cost effective and promising for producing quality thin films over large and non-conductive areas. It is not necessary to operate pressure vessels or other complex equipment. With CBD, it is possible to obtain nanostructured thin films where the process parameters are the choice of precursor chemicals, the solution concentration, the growth temperature, and the growth time [15,16,20]. Many studies have been published about boron-doped ZnO (ZnO:B) using various methods; for example, pyrolysis, radio frequency magnetron sputtering, and ion implantation [41–43]. However, few studies have been published on the properties of ZnO:B manufactured using the chemical bath deposition method, compared with a vast body of literature available on doping ZnO with other elements such as Ag, K, Cu, Sr, Ni, etc. [44–47]. To the best of our knowledge, B-doped ZnO films processed via CBD have been studied in the literature [10]. However, only high B doping concentration (1–3%) effects have been investigated in the study. It is hard to see the electrical measurements results and the effect of substrates (stainless steel, molybdenum, and glass) on the structural, optical, and electrical properties. However, it is hard to follow the details of the electrical properties in the study. Although there are several studies on the films using spray pyrolysis, sol-gel, chemical vapor deposition, radio frequency magnetron sputtering, pulse laser deposition and molecular beam epitaxy methods, as mentioned above, few studies are available on B-doped ZnO films using the chemical bath deposition technique.

Therefore, investigating the effect of doping with boron by the CBD method on the properties of ZnO thin films has been an interesting point, including low doping concentrations (0.1, 0.3, and 0.5%) and high doping concentrations (1.0, 3.0, 5.0, and 7.0%). This study explains how the structural parameters such as grain size, cell distortion degree, Zn-O band length, lattice tension and dislocation density, surface roughness, optical band gap energy and activation energy of ZnO has changed upon boron doping.

## 2. Materials and Methods

*A. Film Synthesis*

Soda-lime glass (SLG) microscopy slides (1 mm × 25 mm × 75 mm) were used as substrates for the deposition of undoped ZnO and ZnO:B nanostructured thin films. The substrates were cleaned with sulfuric acid, diluted with distilled water ($H_2SO_4$: $H_2O$, 1:5 $v/v$) to remove the natural oxide layer ($SiO_2$), then washed with acetone and double-distilled water in an ultrasonic bath and dried at room temperature. Throughout the experiments, all chemical materials used were high-purity reagents purchased from Merck.

The starting solution bath was prepared by using 0.1 M Zn $(NO_3)_2$.$6H_2O$ ($\geq$98%) and an aqueous ammonia solution. The initial pH value was adjusted to ~11. Next, 0.1 M $H_3BO_3$ ($\geq$99.5%) as the boron doping solution was added to the precursor solution with a boron concentration by volume of 0.1, 0.3, 0.5, 1.0, 3.0, 5.0 and 7.0 mL. The films are labeled within the text as follows: S0: undoped ZnO; S1: 0.1%; S2: 0.3%; S3: 0.5%; S4: 1.0%; S5: 3.0%; S6: 5.0%; S7: 7.0%, respectively.

All solutions were mixed on a hot plate for 20 min until they became homogeneous and transparent at room temperature. The cleaned substrates were vertically inserted into the solution bath and then allowed to boil to 366 K for 20 min, and then cooled down in the bath.

The substrates were removed from the bath after 40 min and washed with double-distilled water and dried at ambient atmosphere. An undoped (ZnO) film was deposited under the same conditions in order to serve as reference. After the deposition, all samples were annealed at ambient atmosphere in a furnace (MKF106, Miprolab) in order to remove possible hydroxyl phases at 573 K for 1 h.

*B. Possible Growth Mechanism of ZnO Thin Films*

In the method used for this study, the $[Zn(NH_3)_4]_2$ tetrammine complex was thermally dissolved in water and released $Zn^{2+}$ and $OH^-$ ions into the solution. With the addition of ammonia into the solution, the growth process can be considered dynamically balanced.

Two growth regimens, called homogeneous nucleation and heterogeneous nucleation, are possible during the CBD process. In the homogeneous nucleation mechanism, between the individual ions or molecules, the core ZnO nanoparticles, which are unstable and prone to re-dissolution, combine to form a regular structure. These combined core nanoparticles begin to gather and grow within themselves. These structures can be re-dissolved in a solution before they have a chance to develop into stable particles (nuclei).

On the other hand, in heterogeneous nucleation, this core layer (or individual ions) can be adsorbed onto the substrate. The energy required to create an interface between the core layer and the solid substrate is generally less than that required for homogeneous nucleation in the absence of such an interface. Therefore, heterogeneous nucleation is preferred as an energetic reaction over homogeneous nucleation, and the probability of its occurrence is generally higher than homogeneous nucleation and closer to the required equilibrium conditions [20,48].

The CBD method is based on heating the alkaline zinc salt solution and precipitation of the ZnO nanoparticles into the substrates. In the meantime, when the aqueous ammonia added is added to the zinc nitrate solution, the white precipitate of $Zn(OH)_2$ occurs. Excess ammonia is added to dissolve the $Zn(OH)_2$ precipitate. This can be represented by the following possible reaction equations [49]:

$$Zn(NO_3)_2 + 2NH_4OH \rightarrow Zn(OH)_2(s) + 2NH_4NO_3, \tag{1}$$

$$Zn(OH)_2(s) + NH_4OH_2 \rightarrow (NH_4)ZnO_2^- + H_2O + H^+, \tag{2}$$

When the solution is heated, the ionic product exceeds the solubility limit, and the precipitate condenses on the substrate and begins to form to ZnO nuclei in some of the solution. As a result, the ZnO film is deposited on the substrate by the reaction below:

$$(NH_4)ZnO_2^- + H^+ \rightarrow ZnO + H_4OH, \tag{3}$$

After deposition, the films were subjected to heat treatment at 573 K for 1 h in order to separate possible hydroxyl phases from the structure.

The crystallographic structures of the ZnO and ZnO:B films were characterized by using a X-ray diffractometer (XRD, Rigaku Smartlab X-Ray diffractometer, Tokyo, Japan) using Cu K$\alpha$ radiation ($\lambda$ = 1.54059 Å) in steps of 0.01°·within the 2$\theta$ range of 20–80° at room temperature. The atomic ratios of the elements were determined via X-ray photoelectron spectroscopy (XPS). The XPS measurements were made by a Specs Flex-Mod system (Berlin, Germany) with an Al anode energy of 1486.6 eV monochromatized with single-crystal quartz. The photoelectrons were measured with a 150 mm energy analyzer equipped with 2D CCD detector. Survey measurements were performed with 50 eV pass energy while the core level spectral measurements of Zn, B, O were performed with 20 eV pass energy. Raman spectra of the films were measured using a WITech alpha 300 R Raman spectrometer. The morphological and topological properties of the films were determined by using scanning electron microscopy (SEM, ZEISS EVO-LS10, NTS, Munich, Germany) and atomic force microscopy analyses (AFM, Park System-XE-100E, Suwon, Korea). The scanned area on each sample surface was determined as 5 μm × 5 μm and the scans were performed using the non-contact mode.

The electrical properties of the samples were investigated using the two-point probe method at the temperature range of 300–540 K using a homemade Labview program and a computer-interfaced GW Instek multimeter, GPD 4303S (New Taipei, Taiwan) with high accuracy and a voltage source. The optical absorption and refraction spectra of the pure and B-doped ZnO films were measured using an UV-Vis spectrophotometer at the wavelength range of 200–1100 nm.

## 3. Results and Discussion

### 3.1. Crystallographic Properties of the Pure and B-Doped ZnO Films

The X-ray diffraction patterns were recorded to investigate the effects of boron doping on the structural properties of ZnO thin films. The X-ray diffraction patterns of the films are given in Figure 1. The results showed that all orientations were indexed in accordance with hexagonal phase crystalline ZnO (wurtzite structure space group: P63mc (186); a = 0.3249 nm, c = 0.5206 nm, c/a = 1.6023, JCPDS card no: 005-0664) [50]. The results are in accordance with the literature [51,52].

Figure 1 shows the X-ray diffractograms of the pure ZnO and B-doped ZnO thin films. The most prominent peak is the (002) Bragg reflection. This suggests that the crystallites are preferentially oriented with the c-axis to the substrate plane. Likely due to the preferential orientation of the grains, the diffraction peak (004) could not be observed due to its very weak density, except for the S3 and S5 doped samples. This preferential growth characteristic is associated with the fact that film self-ordering occurs in an order to reduce surface energy during growth [53–55]; see Section 3.4. The film grains, after coalescence, grow mainly in the c-axis direction normal to the substrate surface. In the case of a hexagonal crystal structure, this growth will be in the same orientation due to the most tightly packed structures having the lowest free surface energy in the (002) plane [56]. Therefore, crystallization occurs in this direction. The increase in boron content in the precursor solution was caused by an increase in the peak intensity of the preferential (002) orientation and, as a result, showed that ZnO nanostructures strongly influenced the crystallization process.

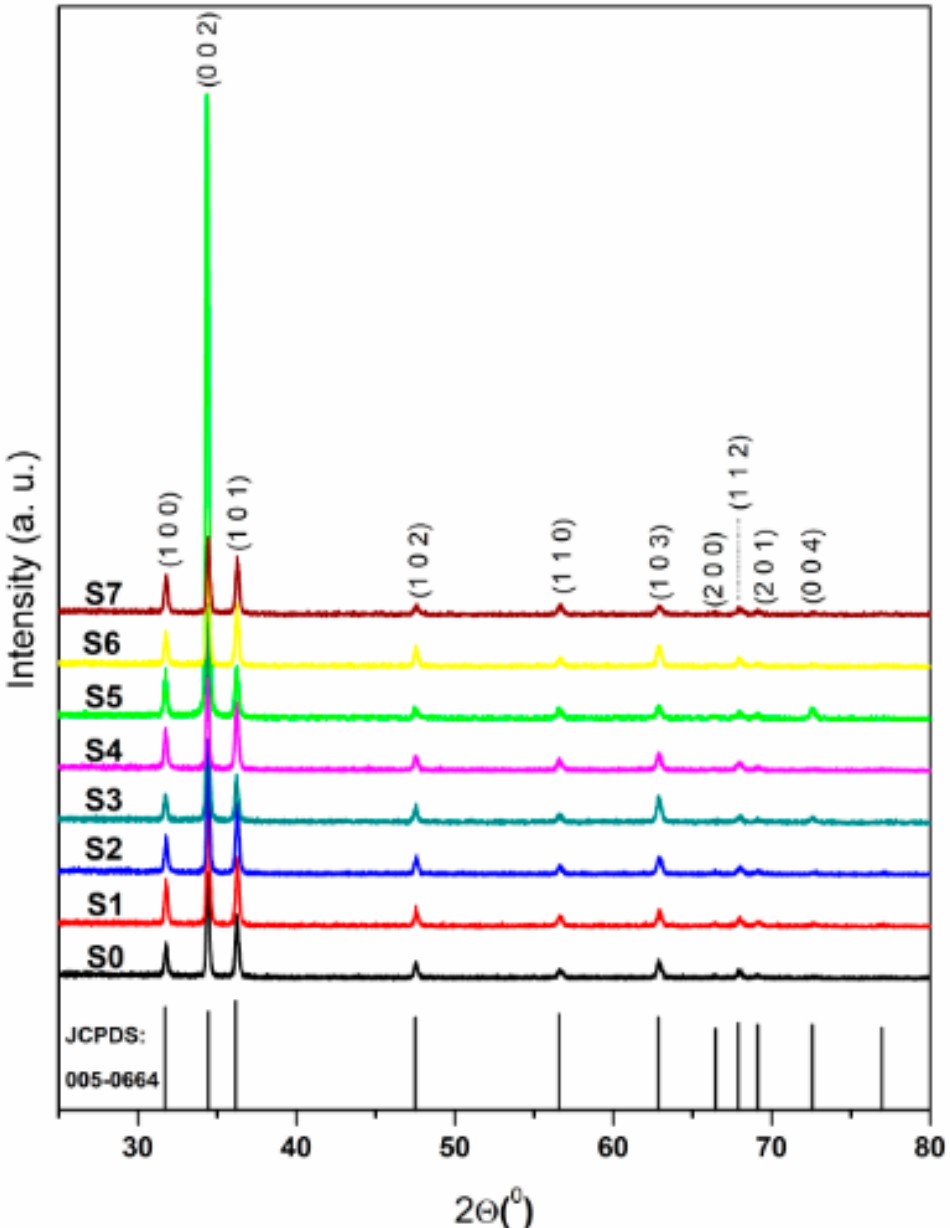

**Figure 1.** The X-ray diffraction patterns of pure ZnO and B-doped ZnO films.

The mean crystallite size of the samples, D, can be calculated using Scherrer's formula as shown in Equation (4) [57]:

$$D = \frac{K\lambda}{\beta_{hkl} \cos \theta} \tag{4}$$

where θ is the value of the angle corresponding to the center of a chosen XRD peak, λ is the wavelength of X-ray radiation and β (hkℓ) represents the integral width from the considered peak of the analyzed sample. For the shape factor, K is taken to be 0.94.

Equations (5) and (6) can be used to find the effects of the doping element on some structural values of the samples such as the micro-strain (ε) and dislocation density (ρ) [58].

$$\varepsilon = \beta \cos \theta / 4 \tag{5}$$

$$\rho = 15\varepsilon / \alpha D \tag{6}$$

where $\alpha$ is the lattice constant. From Table 1, it can be seen that the micro-stress and dislocation density values changed after the doping process, which indicated deterioration in the pristine ZnO lattice.

**Table 1.** Some structural parameters obtained and calculated from the films (D, average crystallite size; $\varepsilon$, micro-strain; $\rho$, dislocation density; d, distance between planes).

| Sample ID | (100) | | (002) | | (101) | | D (nm) | $\varepsilon \times 10^{-4}$ | $\rho \times 10^{15}$ (cm$^{-2}$) |
|---|---|---|---|---|---|---|---|---|---|
| | 2$\theta$ (deg.) | d (Å) | 2$\theta$ (deg.) | d (Å) | 2$\theta$ (deg.) | d (Å) | | | |
| S0 | 31.752 | 2.8159 | 34.404 | 2.6046 | 36.214 | 2.4785 | 38.54 | 9.54 | 1.18 |
| S1 | 31.769 | 2.8144 | 34.419 | 2.6035 | 36.250 | 2.4761 | 39.42 | 9.27 | 1.10 |
| S2 | 31.773 | 2.8141 | 34.415 | 2.6038 | 36.246 | 2.4764 | 36.94 | 10.1 | 1.34 |
| S3 | 31.712 | 2.8193 | 34.400 | 2.6049 | 36.185 | 2.4804 | 37.05 | 9.97 | 1.29 |
| S4 | 31.735 | 2.8173 | 34.382 | 2.6062 | 36.206 | 2.4790 | 37.60 | 9.81 | 1.25 |
| S5 | 31.696 | 2.8207 | 34.341 | 2.6092 | 36.235 | 2.4771 | 40.91 | 9.02 | 1.06 |
| S6 | 31.748 | 2.8162 | 34.393 | 2.6054 | 36.253 | 2.4759 | 37.45 | 9.80 | 1.24 |
| S7 | 31.772 | 2.8141 | 34.418 | 2.6036 | 36.243 | 2.4765 | 34.50 | 10.7 | 1.47 |
| Reference [20] | 31.770 | 2.8143 | 34.422 | 2.6033 | 36.253 | 2.4759 | - | - | - |

Additionally, from Table 1, it can be seen that in samples with different doping ratios, the (100), (002) and (101) characteristic peaks in particular have slightly higher 2θ values than the pristine sample. Such a shift indicates the presence of induced lattice stresses in the structures [59]. As a result of the Bragg formula, various factors such as impurities (dopant atoms), lattice defects, voids, or deformation faults can alter the interplanar distance, as a result of the lattice stress induced in the structure during the deposition process and can cause a shift towards a lower or higher angle [60].

The actual lattice of ZnO deviates from the ideal lattice state by changing the c/a ratio or u value [61]. The experimentally observed c/a ratio is less than ideal, since it was assumed that it would not lead to another phase (zinc blend) [62]. This shows that there is a strong relationship between the c/a ratio and the u parameter when this ratio decreases.

The *u* parameter is an important structural parameter affected by the distortion of tetrahedral angles due to long-region polar interactions. It is calculated by Equation (7), given below. It is also the measure of the displacement of an atom towards its nearest neighbor along the c-axis.

$$u = \frac{a^2}{3c^2} + \frac{1}{4} \tag{7}$$

$$L = \sqrt{\frac{a^2}{3} + \left(\frac{1}{2} - u\right)^2 c^2} \tag{8}$$

The distance between the Zn-O bonds (*L*) depends on the *u* parameter, which can be calculated by Equation (8). The calculated values of *L* are given in Table 2.

In the literature, the ionic radius of $O^{-2}$ is 1.38 Å and the ionic radius of $Zn^{2+}$ is 0.83 Å [63]. Therefore, the theoretical length (L) of the Zn-O bond is 2.21 Å. The values obtained for our samples have very near values to the theoretical estimated value of the L parameter, which can indicate structural defects such as vacancies of oxygen and zinc [64]. The lattice distortion parameter, R can be calculated using the Equation (9).

$$R = (2a((2/3)/c)^{1/2} \tag{9}$$

R = 1 means that there is no distortion in the structure [65].

It is clear to see in Figure 2 that the lattice constants and L value increase in value up to the 3% doping concentration. Smaller lattice constants were obtained from an ideal

wurtzite crystal structure. It has been reported that this may be related to the instability of the crystal lattice structure, point defects such as zinc antisites, and defects such as oxygen space defects and hetero-epitaxial dislocations [66,67]. The structural parameters from the analysis of the X-ray diffraction are summarized in Table 2.

**Table 2.** Structural parameters of the pure and B-doped ZnO thin films.

| Sample | Lattice Constant | | | Length of the Zn-O Bond | Lattice Distortion Degree | 2θ | |
|--------|--------|--------|---------|--------|--------|--------|--------|
| | a (Å) | c (Å) | c/a | L (Å) | R | (100) | (002) |
| S0 | 3.2503 | 5.2074 | 1.60212 | 2.284 | 1.0193 | 31.752 | 34.404 |
| S1 | 3.2486 | 5.2052 | 1.60227 | 2.282 | 1.0192 | 31.769 | 34.419 |
| S2 | 3.2482 | 5.2058 | 1.60265 | 2.283 | 1.0189 | 31.773 | 34.415 |
| S3 | 3.2543 | 5.2080 | 1.60033 | 2.286 | 1.0204 | 31.712 | 34.400 |
| S4 | 3.2520 | 5.2106 | 1.60227 | 2.285 | 1.0192 | 31.735 | 34.382 |
| S5 | 3.2559 | 5.2167 | 1.60221 | 2.288 | 1.0192 | 31.696 | 34.341 |
| S6 | 3.2507 | 5.2090 | 1.60242 | 2.284 | 1.0191 | 31.748 | 34.393 |
| S7 | 3.2483 | 5.2054 | 1.60247 | 2.283 | 1.0190 | 31.772 | 34.418 |

Figure 2 shows that the R parameter rises sharply at the 0.5% doping level and exhibits a linear behavior at values beyond this percentage. Compared with the film without any additive, boron-doped ZnO samples have high c/a and L values, while they have low R values at a higher boron content. While the lattice parameters (a, c) increased at values below 3% boron concentration, the parameters decreased at higher boron concentration than 3%. The parameter L, which stands for the Zn-O bond length, shows the same trend. The ionic radius of $Zn^{2+}$ and $B^{3+}$ ions is 0.83 Å and 0.23 Å, respectively. Therefore, a decrease in the lattice parameter is expected as the concentration of boron increases. The lattice parameter a decreased for S1 and S2 and increased for S3. It seems that the a parameter value decreases for high doping levels except for S5. It was also mentioned in the literature that the crystallographic cell parameters of the ZnO films change with B doping. However, the change in the cell parameters is not linear with an increasing B concentration [51,52]. The calculated cell parameters in the present study are in agreement with [51].

### 3.2. X-ray Photoelectron Spectroscopy Study

Figure 3 shows a representative XPS survey spectrum of the boron-doped ZnO thin films. As seen in the figure, only XPS peaks belong to the Zn, O, B, and C elements in all thin films. The actual doping obtained from the XPS and the nominal values of B content within the films are also presented in Figure 3 (inset, left). The plot shows a linear variation in the nominal values and the actual doping. Due to the fact that the B 1 s peak intensity is quite low and difficult to follow in the survey spectra for the reader, the peak with higher magnification is presented in Figure 3 (inset, right).

The XPS measurements have been used to obtain the atomic percentages of the elements. The results given below show the atomic percentages of Zn, O, and B atoms in the boron-doped ZnO thin films, along with the nominal content of boron (Table 3). As seen in Table 3, as the boron doping content (*v/v* %) increased, the actual content of B in the films also increased homogeneously. The maximum atomic percentage of B obtained by XPS was 6.33% for S7.

**Table 3.** The nominal content of B (*v/v* and at. %) and the actual content of elemental Zn, O, and B in the pure and B-doped ZnO thin films.

| Sample | Nominal Content | | Actual Content | | | $O_V/O_{M-O}$ |
|---|---|---|---|---|---|---|
| | *v/v* (%) | at% | Zn% | O% | B% | |
| S0 | 0.0 | 0.0 | 35.37 | 64.63 | - | 0.90 |
| S1 | 0.1 | 0.13 | 33.29 | 65.89 | 0.82 | 0.86 |
| S2 | 0.3 | 0.38 | 39.97 | 58.73 | 1.31 | 0.48 |
| S3 | 0.5 | 0.63 | 35.75 | 62.59 | 1.66 | 0.58 |
| S4 | 1.0 | 1.25 | 39.24 | 58.89 | 1.88 | 0.42 |
| S5 | 3.0 | 3.74 | 38.50 | 58.22 | 3.28 | 0.54 |
| S6 | 5.0 | 6.21 | 32.68 | 61.36 | 5.96 | 0.58 |
| S7 | 7.0 | 8.64 | 35.70 | 57.97 | 6.33 | 0.62 |

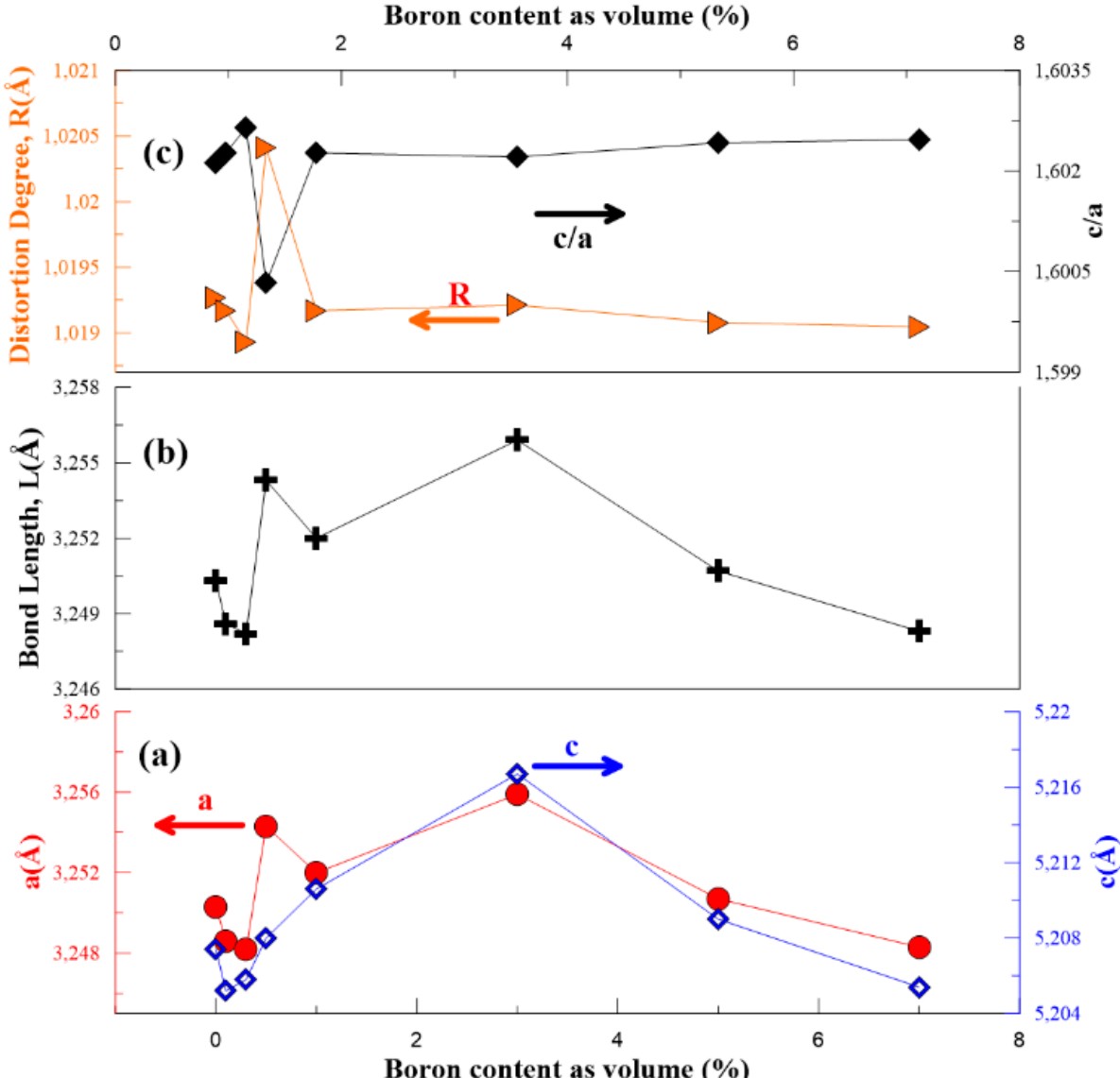

**Figure 2.** The variation in the (**a**) lattice parameters a and c, (**b**) Zn-O bond length, and (**c**) degree of lattice distortion and ratio as a function of boron content by volume.

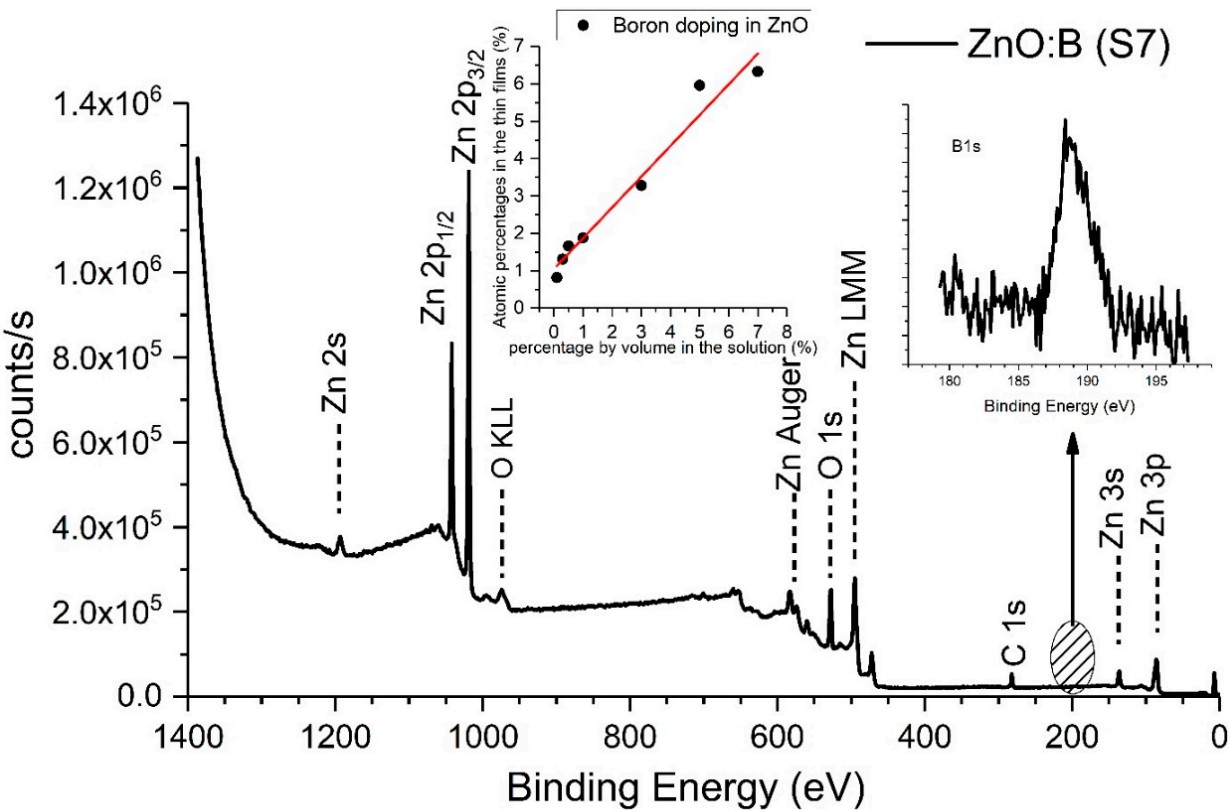

**Figure 3.** A representative XPS survey spectrum of the boron-doped ZnO thin films. The inset shows (**left**) the actual dopings obtained from the XPS and the nominal values and (**right**) a magnification of the spectrum with the B 1 s peak.

For further investigation of the chemical effects of doping, the Zn, O, and B high-resolution core spectra were investigated for all thin films, as seen in Figure 4. The binding energy of the Zn $2p_{1/2}$ and Zn $2p_{3/2}$ core spectra is 1022.0 eV and 1045.0 eV, respectively. These binding energies confirm the metal-oxide formation. There is no shift observed in the Zn core level spectra in the boron-doped ZnO thin films, indicating no deterioration of the metal-oxide bond by boron doping.

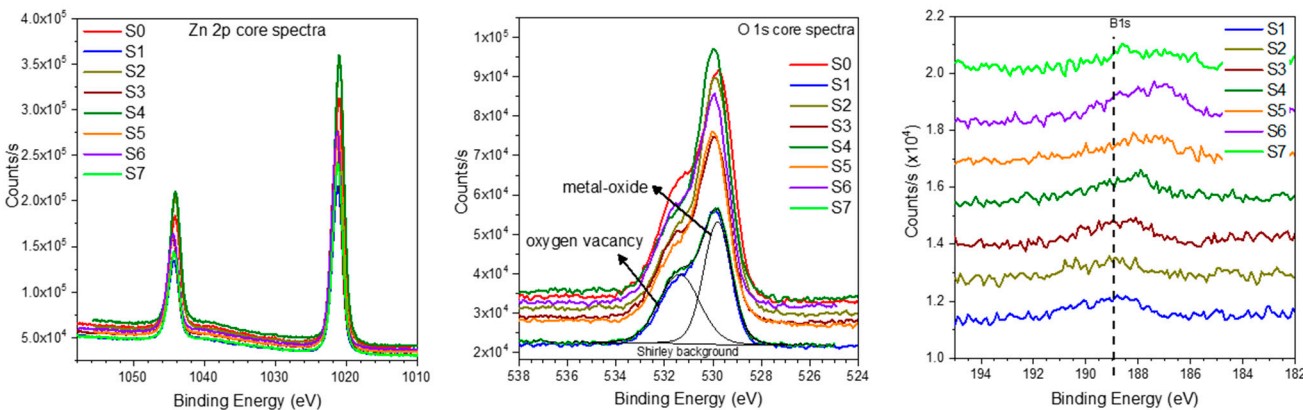

**Figure 4.** High-resolution XPS spectra of pure and B-doped ZnO films.

The O 1s high-resolution spectra for all the thin films show one main peak, and the shoulder appears with lower intensities at higher energies. We show one representative least square fit made to the O 1s spectrum for the S7 sample in Figure 4. The peak that appeared on the lower energy side is commonly attributed to the oxygen metal-oxide bond

in the stoichiometric ZnO lattice, while the peak on the higher energy side is due to the metal ions in an oxygen-deficient environment (frequently termed oxygen vacancy (V)) in the lattice [68]. The ratio of oxygen vacancy to the metal-oxide bond ($O_v/O_{M-O}$) decreased in the boron-doped ZnO thin films, as indicated in Table 3, compared with the pure ZnO. Apparently, the boron helps to structure the ZnO towards a more stoichiometric phase. The high-resolution XPS spectra in Figure 4 show a shift of the B 1s intensity towards lower binding energy upon increasing the B doping level, indicating that boron has the tendency to form a metallic phase in ZnO.

### 3.3. Raman Spectroscopy Study

The optical Raman spectra of the films look virtually identical, showing that there are no substantial changes in the vibration structure of the ZnO films upon doping with boron. The Raman peaks can be assigned to the optical phonon modes of the wurtzite crystal structure of ZnO. As can be seen in Figure 5, the band at about 438 cm$^{-1}$ is attributed to the $E_2$ (high) mode of the ZnO single crystalline hexagonal cohort structure of all samples.

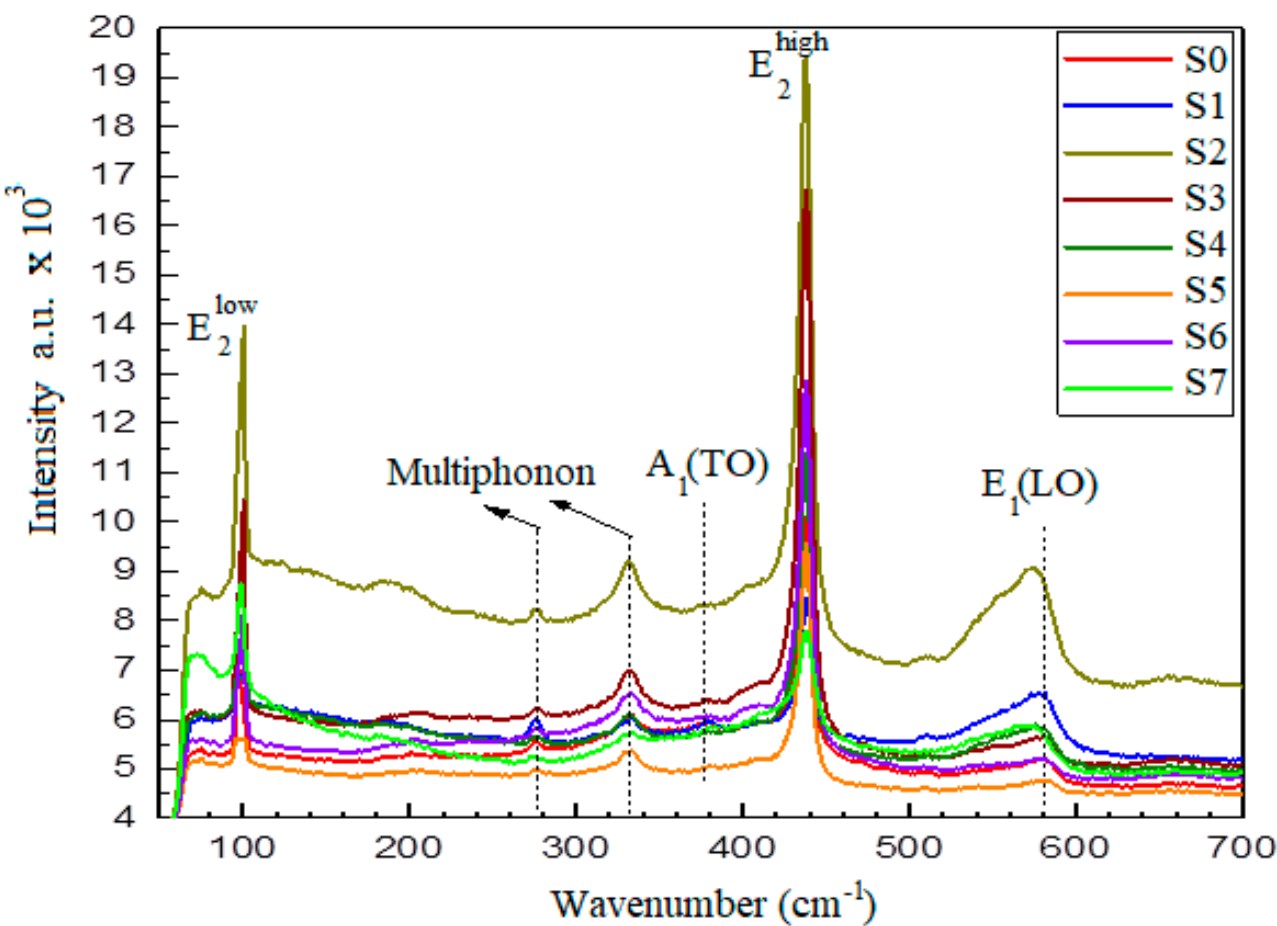

**Figure 5.** Raman spectra of the pure and B doped ZnO films.

The peaks at both 278 and 336 cm$^{-1}$ are assigned to the $A_1$ symmetry modes and multiple-phonon scattering processes or [$E_2$ (high)-$E_2$ (low)]. The peak at 579 cm$^{-1}$ is assigned to the $E_1$ (LO) mode and is related to host defects such as $V_O$ and $Zn_i$. The peak at 381 cm$^{-1}$ corresponds to the $A_1$ (TO) modes. As a result of that, no Raman peak of boron appeared in the spectrum of the B-doped ZnO nanostructures.

### 3.4. Scanning Electron Microscopy Study

Figure 6 shows the scanning electron micrographs of the plain view of ZnO and ZnO:B films at a magnification of 1.00 and 10.00 K X. It is obvious that both pure and B-doped ZnO films are well covered on the substrates. It is clearly seen in Figure 6 that the undoped sample (S0) consists of dense, continuous, and hexagonal polycrystalline nano-rods, which are perpendicular to the substrate's surface.

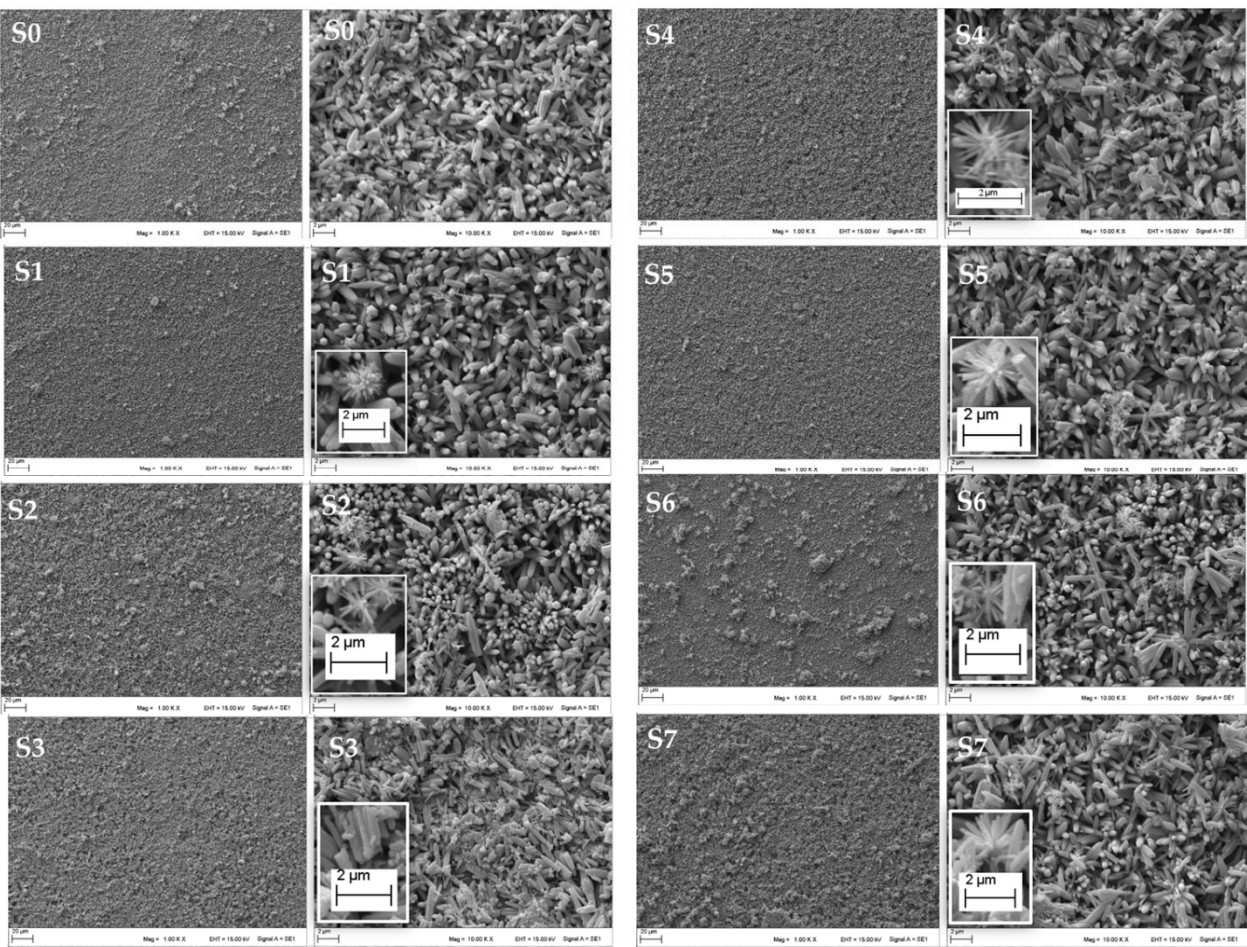

**Figure 6.** SEM micrographs of the pure ZnO (S0) and B-doped ZnO films (S1: 0.1%; S2: 0.3%; S3: 0.5%; S4: 1.0%; S5: 3.0%; S6: 5.0%; S7: 7.0%).

According to the crystal growth mechanism, the growth kinetics of crystallites are determined by the orientation of the crystal. Depending on their orientation, growth competition may begin between neighboring crystals. Faster-growing crystals tend to grow over slower-growing ones. When this growth progresses towards the formation of the same type of crystal faces, they form a free surface. This mode of growth represents a choice of orientation and leads to competitive growth [21–28]. The preferred orientation for ZnO micro-scale rods is (002), along the c-axis. Generally, II–VI binary compound semiconductors crystallize in a cubic zinc blend or hexagonal wurtzite symmetry because of a thermodynamically stable phase in which each anion is surrounded by four cations at the corners of a tetrahedron [1]. This tetrahedral coordination is inherent in the sp3 covalent bond.

The SEM micrographs of the films shown in Figure 6 are labeled from S1 to S7 for the boron-doped ZnO films in plain view, respectively. It is obvious that the boron doping affects the surface morphology of ZnO, and the vertical alignment of the doped structures is lower than that of the pristine one.

The atomic radius of boron is about 80 pm, while that of Zn is 139 pm. The electronegativity of these two elements is 2.04 and 1.65, respectively. Modifications of the morphologies can be related to the thermodynamically stable growth mechanism of the crystal faces of ZnO, the different atomic radii affecting the free surfaces, and the electronegativity of the dopant. Depending on the boron additive ratio (Figure 6S1–S7), it can be seen that independent hexagonal nano-rod structures come together to form dandelion flower-like structures, which are shown in the inset. In Figure 6, it can be seen that with an increasing boron concentration, dandelion flower-like structures are formed on the surface and the micro-scale rods in different directions show a random distribution and accumulation on the surface. It has been observed that these micro-scale rods are formed in smaller sizes as the concentration of boron increases and covers the entire surface in dense clusters. Upon boron doping, the rods, which create the dandelion-like flower structure have been grown. As a result of this phenomenon, it can be said that doping with boron significantly affects the growth behavior of ZnO.

Consequently, it can be understood that the morphological changes in ZnO are related to the different atomic radius and electronegativity of the additive, and the unstable growth mechanism.

### 3.5. Atomic Force Microscopy Study

AFM was used to characterize the surface roughness of different boron-doped (0.1%, 0.3%, 0.5%, 1%, 3%, 5%, and 7%) ZnO micro-structures. The average of the surface roughness (Ra) analyses was calculated as the standard deviation of all the height values within a given area (Table 4).

**Table 4.** Values of the root mean square roughness (Rq), the average roughness (Ra), and 10-point average roughness (Rz) of the films.

| Samples | Rq (nm) | Ra (nm) | Rz (nm) |
|---------|---------|---------|---------|
| S0 | 48 | 112 | 170 |
| S1 | 272 | 216 | 431 |
| S2 | 158 | 222 | 170 |
| S3 | 136 | 209 | 712 |
| S4 | 252 | 213 | 793 |
| S5 | 365 | 251 | 938 |
| S6 | 298 | 244 | 913 |
| S7 | 203 | 317 | 918 |

Figure 7 (right side of each sample) shows three-dimensional AFM images of the pristine ZnO thin film and the different boron-doped ZnO samples. The values of the root mean square roughness (Rq), the average roughness (Ra), and the 10-point average roughness (Rz) of ZnO are given in Table 4. It can be seen that the labeled S5 film shows average surface roughness (Ra).

### 3.6. Optical Properties

The optical absorbance of the boron-doped ZnO thin films synthesized by the CBD method on glass substrates was measured by using an UV spectrophotometer at a wavelength range of 200–1000 nm.

The optical band gap energies of the ZnO and boron doped ZnO thin films were calculated as 3.28, 3.18, 3.18, 3.20, 3.19, 3.21, 3.22 and 3.22 eV, respectively. Ahmad et al. also calculated the optical band gap energies and found that the $E_g$ of pure ZnO decreases from 3.37 eV to 3.17 eV upon B doping [69]. This shows that increasing the doping concentration results in decreased band gap values in the structures (Figure 8).

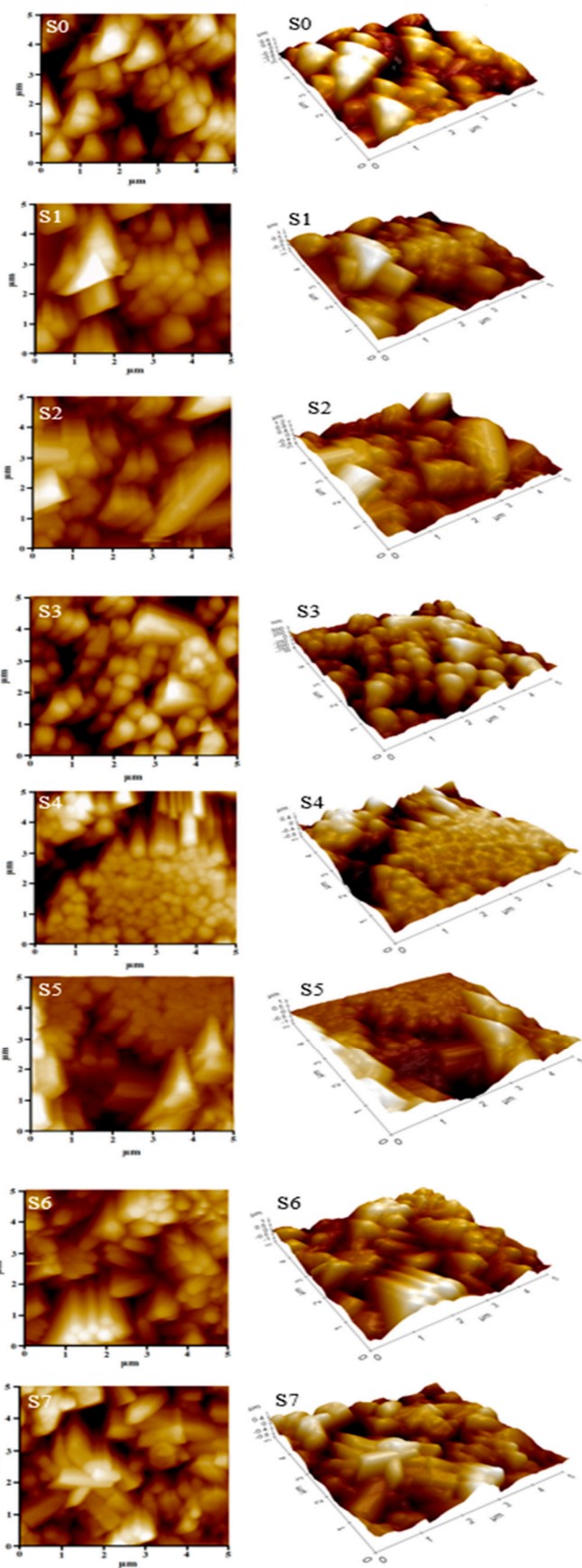

**Figure 7.** AFM images of the pure and B doped ZnO films. Right column shows three-dimensional AFM images and left column shows two-dimensional AFM images.

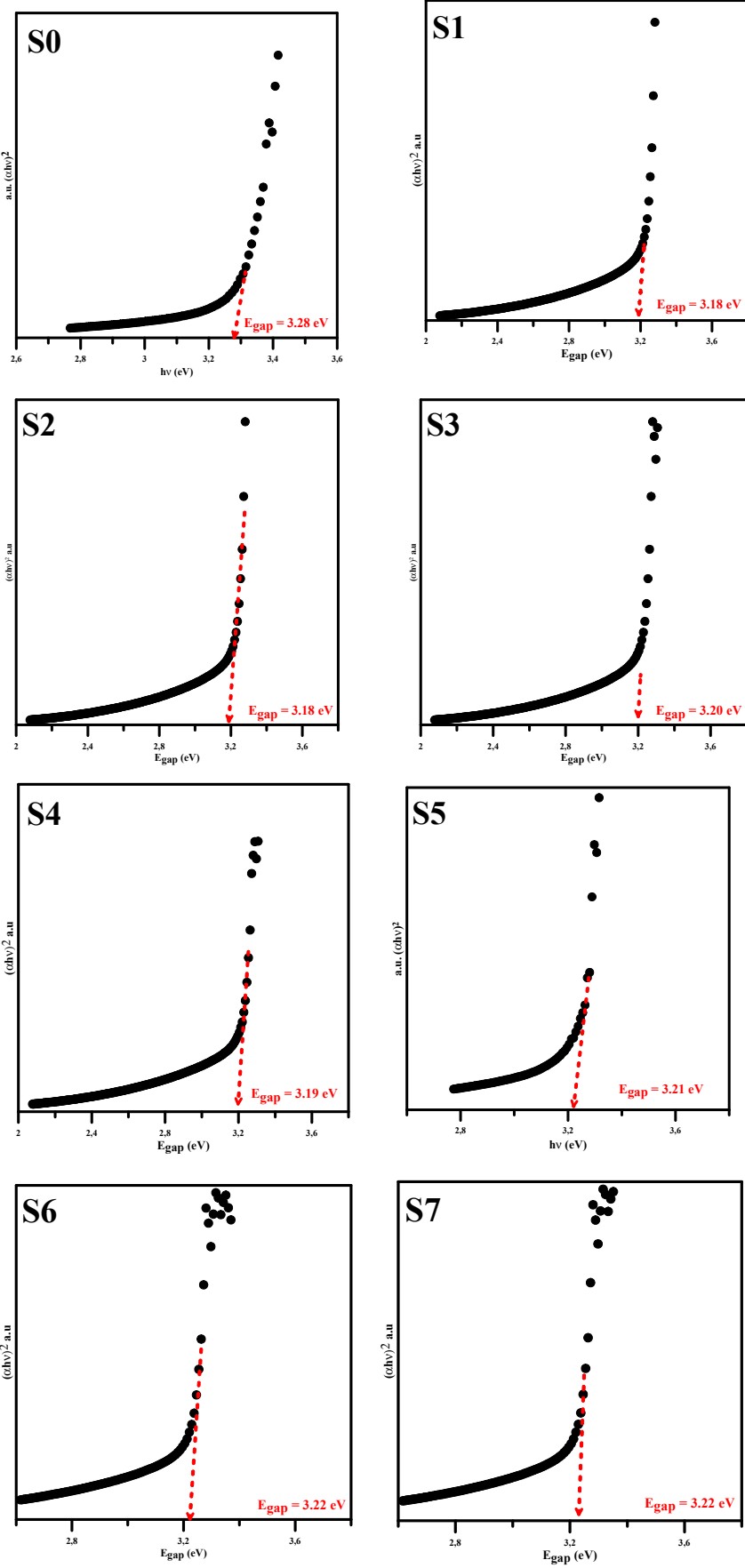

**Figure 8.** The optical band gap energy of the pure and boron doped ZnO thin films.

In order to make more detailed analysis of the reflectance spectra of the films, the process of deconvolution was performed in the wavelength range of 410-370 nm. The differential reflectance dR/dλ versus λ is also shown in Figure 9. The corresponding $E_g$ values can be estimated for these films by using the average of the maximum values of these plots. The calculated values are 3.22, 3.21, 3.21, 3.22, 3.20, 3.22, 3.22 and 3.25 eV, respectively.

### 3.7. Electrical Properties

In order to find the electrical activation energies of the impurity levels, the conductivity–temperature characteristics of all the samples were examined by the two-point probe method in the temperature range of 300–723 K with a GW Instek 8261A precision multimeter and GW Instek 4303S voltage source interfaced with a computer by a LabVIEW program. Ohmic contacts were provided by silver pasted on the electrodes. Generally, pristine ZnO thin film nanostructures have an n-type conduction due to the intrinsic defects, which are ascribed to native defects such as the Zn interstitial atoms and the oxygen vacancies [70].

According to the solid-state theory of semiconductors, in the case of a semiconductor with one or more impurity levels, the temperature dependence of dark electrical resistance is given by Equation (10):

$$R(T) = R_0 e^{E_g/2kT} + \sum_{i=1}^{n} R'_{0,i} e^{\Delta_i/kT} \qquad (10)$$

where $R_0$ and $R'_{0,i}$ are constants, $E_g$ is the thermal band gap energy, $E_i$ is the impurity level ionization energy, k is the Boltzmann constant, and T is the absolute temperature in Kelvin. In the low-temperature region, the total conductivity of the semiconductor samples is dominated by charge carriers generated by the ionization (extrinsic conductivity) of the impurity levels and hence the second term dominates at room temperature. Particularly at high temperatures, the electronic transitions from band to band are responsible for the temperature dependence of the conductivity. Under these conditions, charge carriers obtain sufficient thermal energy to make an inter-band transition [71]. According to the information given above, the two terms appearing in Equation (10) may be assumed independently in the corresponding temperature intervals. Therefore, in the graph of Ln(R) versus 1000/T (Figure 10), a series of linear trends appear, showing the forbidden band gap and a set of impurity levels (band gap states) in the forbidden band gap. However, for a semiconductor with a wide forbidden band gap such as ZnO, the conductivity is dominated by charge carriers generated by ionization of the impurity levels in the studied temperature range.

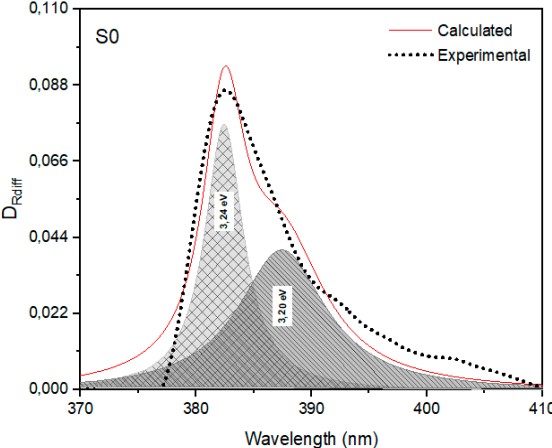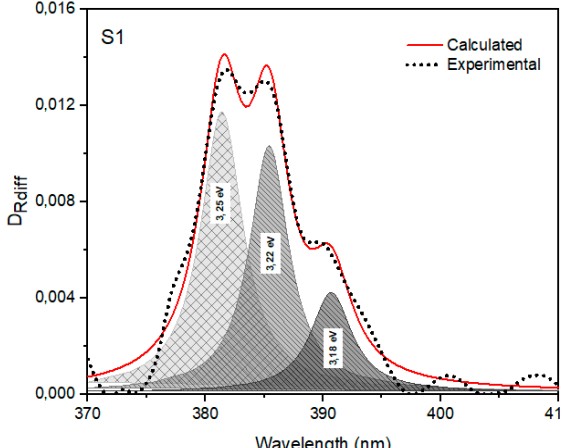

**Figure 9.** *Cont.*

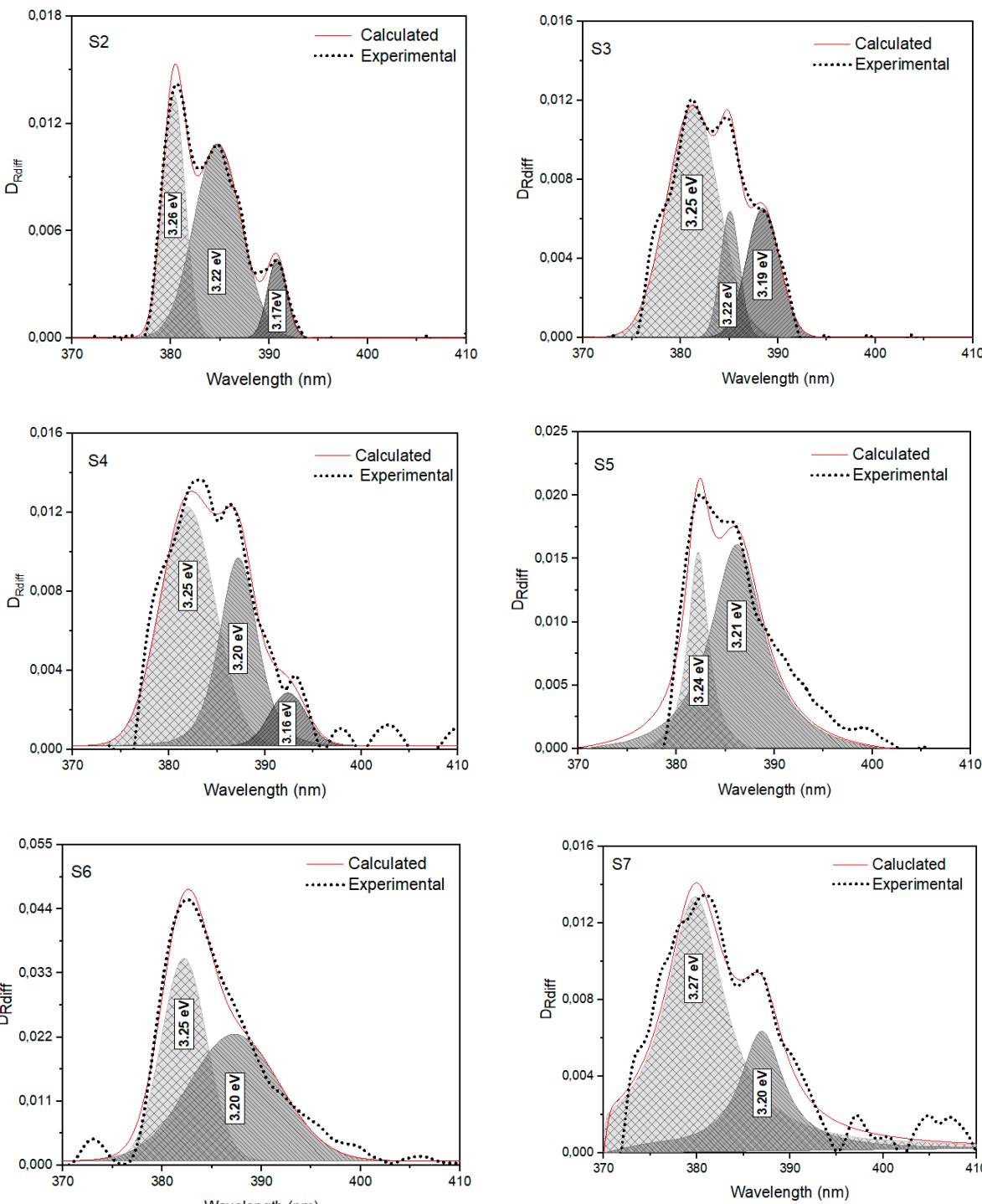

**Figure 9.** Deconvolution of the reflectance spectra for the pure and boron doped ZnO films.

Therefore, the second term in Equation (11) is dominant in RT dependency. Upon increasing the temperature in the extrinsic conductivity zone, it follows the form of a decrease in electrical resistance (corresponding to a lower temperature range).

$$T = \sum_{i=1}^{n} R'_{0,i} \exp\left(\frac{\Delta E_i}{kT}\right) \tag{11}$$

As seen in Figure 10, there are two linear regions; two impurity levels can be calculated for each sample. Equation (12) can easily be written as:

$$\Delta E = k \frac{d(lnR(T))}{d\left(\frac{1}{T}\right)} \tag{12}$$

The impurity levels' ionization energy values were obtained from the slopes of the *Ln* (*R*) vs. 1000/*T* graphs. The obtained impurity levels' ionization energy (Δ*E*) values are given Table 5.

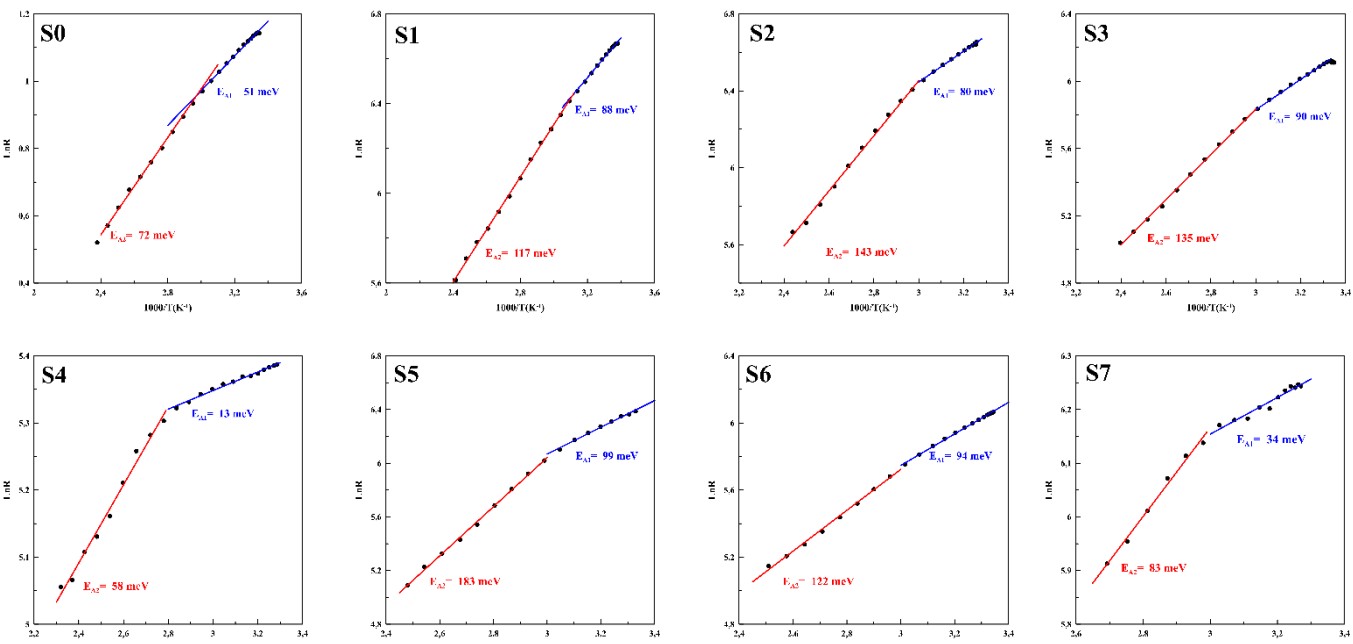

**Figure 10.** Ln (R) vs. 1000/*T* graphs of the films within the temperature range of 300–723 K.

**Table 5.** Activation energies of pure and B doped ZnO thin films for both low- and high-temperature regions.

| Samples | Low-Temperature Region 273–348 K | High-Temperature Region 348–523 K |
|---|---|---|
| | Activation Energy-1 (AE₁) (eV) | Activation Energy-2 (AE₂) (eV) |
| S0 | 0.051 | 0.072 |
| S1 | 0.088 | 0.117 |
| S2 | 0.080 | 0.143 |
| S3 | 0.090 | 0.135 |
| S4 | 0.013 | 0.058 |
| S5 | 0.099 | 0.183 |
| S6 | 0.094 | 0.122 |
| S7 | 0.034 | 0.083 |

Sunanda C. Yadav et al. determined two activation energy values by taking the current–voltage measurements of the samples in the 290–575 K frequency range. The researchers reported that they found values in the low temperature region between 0.22 and 0.93 eV and activation energy values in the high temperature region between 0.17 and 0.66 eV. In the high-temperature region, conduction is attributed to intrinsic defects

caused by thermal fluctuations. The high value of activation energy is mainly determined by intrinsic defects and is therefore called intrinsic conduction [72].

In Figure 9, the graph shows two net conduction zones, one at low temperature and the other at high temperature. The samples also show the feature of decreasing resistivity with increasing temperature, which demonstrates the semiconductor behavior. At relatively high temperatures, the adsorbed oxygen molecules are desorbed from the thin film's surface [20–22], so the potential barrier at the grain boundaries is reduced, making it easier for electrons to cross the grain boundaries. It also affects the increase in donor density due to thermal stimulation.

The activation energy values, which are calculated using Equation (12) for two different temperature regions are given in Table 5. In the relatively low temperature region of conduction, the decrease in resistivity may be due to an increase in the mobility of the charge carriers. Therefore, small thermal energy values are not sufficient for these charge carriers to contribute to conductivity.

As a result, the activation energy depends on the donor carrier concentration and the impurities' energy levels. An increase in donor carrier concentration brings the Fermi level up in the energy gap and results in a decrease in the activation energy [73]. It is assumed that the activation energy is increased as a result of doping defects. It is seen that the activation energy values found in this study are compatible with the literature values.

## 4. Conclusions

In this work, ZnO and B-doped (0.1, 0.3, 0.5, 1.0, 3.0, 5.0, 7.0 $v/v$%) ZnO films were deposited on glass substrates using the chemical bath deposition technique. The effects of the B concentration within the ZnO host on the crystallographic structure, morphology, optical, and electrical properties were investigated in detail. The ZnO and ZnO:B films were obtained with a wurtzite structure and were well crystallized. The B atoms were well incorporated into the ZnO and the preferred orientation of (0 0 2) did not change with an increase in the B doping concentration. The crystallite size of ZnO (38.54 nm) decreased upon doping and the lowest crystal size was calculated as 34.50 nm for the film with the highest B concentration (S7). It was concluded that B doping does not cause any deterioration within the ZnO, for which the binding energy of Zn $2p_{1/2}$ (1022.0 eV) and Zn $2p_{3/2}$ (1045.0 eV) peaks did not shift with increasing B content. The $O_v/O_{M-O}$ ratio of ZnO decreased with B doping, which supports the more stoichiometric phase of ZnO. The ratio of ZnO was calculated as 0.90, while the lowest ratio was 0.42 for the film containing 1.0% B content (S4). The shift in the B 1s peak towards a lower binding energy with an increase in B content indicated that B had the tendency to form a metallic phase in ZnO. The Raman spectra of the ZnO did not change significantly with an increase in B content, and no Raman peaks of boron were observed for the ZnO:B films. The SEM micrographs show that undoped ZnO polycrystalline nano-rods show hexagonal phase top facets with an average diameter and length of about 400 nm and 2000 nm, respectively. The rods, which creates the dandelion-like flower structure, have grown upon B doping. The optical band gap of ZnO (3.28 eV) decreased with B doping. The band gap of the ZnO:B film varied between 3.18–3.22 eV. The Ea of the ZnO was calculated as 0.051 eV, whereas that of the ZnO:B film containing 1.0% B (S4) was calculated as 0.013 eV in the low-temperature region (273–348 K), and 0.072 eV and 0.183 eV in the high-temperature region (348–523 K), respectively.

**Author Contributions:** Conceptualization, S.E. and S.Ç.; methodology, S.E., A.B. and S.Ç.; XRD analysis, S.Y., XPS and Raman analysis, A.E.K. and E.G.; AFM analysis, E.H. and K.O.; all the other aspects such as investigation, resources, writing—original draft preparation, writing—review and editing, visualization, supervision, project administration, and funding acquisition were conducted by all authors. All authors have read and agreed to the published version of the manuscript.

**Funding:** This work was supported by the Scientific Research Commission (BAP) of Mersin University (Project No: 2018-2-AP1-2962, 2018-3-AP5-3091). Moreover, Artur Braun is grateful for the support of project No. 200021E-189455 of the Flanders/Swiss Lead Agency Programme.

**Institutional Review Board Statement:** Not applicable.

**Informed Consent Statement:** Not applicable.

**Data Availability Statement:** All data re contained within the manuscript or are available to be shared upon request (please contact the corresponding author).

**Conflicts of Interest:** The authors have no conflict of interest to declare that are relevant to the content of this article.

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
