# Peer review of "Solution-Processable Growth and Characterization of Dandelion-like ZnO:B Microflower Structures"

_crystals, doi:10.3390/cryst12010011_

Round 1

Reviewer 1 Report

The author present Intrinsic and dandelion-like microflower nano rod structures of boron doped ZnO thin film were synthesized through the chemical bath deposition (CBD) technique. The boron doped ZnO film was well characterized with various analytic tools to investigate the structural parameters such as grain size, cell distortion degree, Zn-O band length, lattice tension and dislocation density, surface roughness, Boron’s binding energy, ionization energies of purity levels and optical band gaps. Nevertheless, the purpose of this work is still not very clear and undefined. Here the reviewer suggests a few questions on the report.  

1) What is the advantage of CBD technique to make boron doped ZnO thin film in terms of performance or structural stability?  

2) The author displayed the as-synthesized boron doped ZnO film and tried to explain the growth mechanism of ZnO thin films. In fact, there are tons of researches on the growth mechanism of ZnO thin films. In addition, the proposed mechanism was not well supported by experimental result. What’s the purpose of this work and how could those works be supported?

3) Author claimed that the boron doping affects the surface morphology of ZnO and the vertical alignment of the doped structures. However, in terms of SEM images (figure 1), it’s not easy to tell such difference in morphology among ZnO films treated/un-treated with boron. Could you please explain on more about that?  If it is true, how does the boron doping affect to the morphology of ZnO film? Is there any evidence to support such claim?  

4) Most of data are not analyzed based on relevant evidence but just displayed.

5) As asked above, the purpose of this paper is not clear. What’s the objective of this work?

Reviewer believes above questions and suggestions would be helpful for the improvement of the paper quality.

Reviewer 2 Report

The authors describe the synthesis of dandelion-like microflower nano rod structures of Boron doped ZnO thin films in this manuscript. It is interesting research. However, in my opinion, the authors show just experimental data, not new discovery or theory. I can’t find novelty in this manuscript. The authors should summarize the experimental data and make a conclusion. After that, re-submission is recommended. Additionally, the resolution of some images too low.  

Author Response

response to reviewer 2

Reviewer 3 Report

Dear Authors, in your manuscript, the following points should be added/changed to further improve:

  1. Abstract: Please explain the abbreviation „CBD”.
  2. Abstract: Please indicate the range of dopant content in the samples received (Boron doped ZnO).
  3. Introductions: In describing the state of the art, I suggest you consider the article “Enhanced sensing performance of ZnO nanostructures-based gas sensors: A review” DOI:10.1016/j.egyr.2019.08.070
  4. Introductions: I have a comment on the sentence “There is a wide variety of fabrication methods for ZnO thin films such as molecular 69 beam epitaxy (MBE), pulsed laser deposition (PLD), atomic layer deposition (ALD), radio 70 frequency (RF) magnetron sputtering, chemical bath deposition (CBD), hydrothermal synthesis, sol-gel spin coating and sol-gel dip coating [14-16, 26-34].” I also suggest including microwave synthesis (DOI:10.3390/nano10061086).
  5. Introductions: I have a comment on the sentence “Few studies have been published on the properties of Boron-doped ZnO (ZnO: B) using the chemcal bath deposition method, compared to a vast number of literature available on doping of ZnO with other elements [10].” I also suggest mentioning other works that have used other methods to obtain Boron-doped ZnO,
  6. Materials and Methods: Please specify the size of the "Soda-lime glass (SLG) microscope slides" used.
  7. Materials and Methods: Please explain to me what "pro analysis quality" means.
  8. Materials and Methods: Please specify the manufacturer of the reagents used.
  9. Materials and Methods: Please provide information on the furnace used (model, manufacturer). Whether the samples were heated in air?
  10. Materials and Methods: I have a comment on the sentence “0.1 M H3BO3 as the boron doping solution was added to the precursor solution at the volume of 0.1, 0.3, 0.5, 1.0, 3.0, 5.0 and 7.0 %. The films were labelled within the text as S0: undoped ZnO and S1:(0.1%), S2:(0.3%), S3:(0.5%); S4: (1%); S5: (3.0%); S6: (5.0%) and S7: (7.0 %) B doped ZnO respectively.” Please explain to me what percentage of doping this is (atomic or mass)? What were the volumes of the solutions mixed together?
  11. Does the "Results" section also include discussions?
  12. Results: Please correctly describe the name of figure 1. What does "S0, S1, S2, S3, S4, S5, S6, S7" mean? I note to the authors that there are no sem images labeled “Figs. (1(A)-(H))”.
  13. Results: Please also include the nominal dopant contents in Table 2.
  14. Crystallographic properties of the pure and B-doped ZnO films: Were the obtained ZnO rods monocrystalline ? Please compare the SEM results (Fig. 1) with the average crystallite size results obtained. Did the authors convert the actual dopant content against the formula for nominal dopant content? How can you explain the obtained difference in doping values (nominal vs. actual).
  15. Crystallographic properties of the pure and B-doped ZnO films: Why the authors named the results of average crystallite size as grain size (Table 3)?
  16. In my opinion, the authors should create a "results and discussion" section.
  17. The conclusion section is missing.

Round 2

Reviewer 2 Report

As the authors said, this manuscript was focused on the crystallographic structure, morphology, and physical properties of Boron doped ZnO films, which synthesized by chemical bath deposition technique. However, in my thought, the author should suggest the concreate relation between the crystallographic structure and physical properties. This is the weak point of this manuscript. I recommend TEM or STEM investigation of Boron doped ZnO films. Furthermore, the SEM and AFM image should have high resolution. And English language and style should be checked before the publication.

Reviewer 3 Report

1. Abstract: Please indicate the range of dopant content in the samples received (Boron doped ZnO).

  • The dopant concentrations are inserted in the abstract. “The boron concentrations are 0.1, 0.3, 0.5, 1.0, 3.0, 5.0, and 7.0% as volume.”

Reply: Please explain to me how the author calculated the dopant value as a volume percentage. This is the first time I have come across a volume percentage of an dopant. General comment: % can be applied to dimensionless quantity.

2. Introductions: In describing the state of the art, I suggest you consider the article “Enhanced sensing performance of ZnO nanostructures-based gas sensors: A review” DOI:10.1016/j.egyr.2019.08.070

  • It is included in the text and the reference list.

Reply: The authors wrote in the abstract “Consequently, it can be interpret these structures may find some application areas such as in sensors of gases and solar cells.” I pointed the authors to a review article where they could check if their material was new for potential applications as “ZnO nanostructures-based gas sensors”. Please answer the question whether there are already reports of gas sensors based on Boron doped ZnO.

3. Materials and Methods: I have a comment on the sentence “0.1 M H3BO3 as the boron doping solution was added to the precursor solution at the volume of 0.1, 0.3, 0.5, 1.0, 3.0, 5.0 and 7.0 %. The films were labelled within the text as S0: undoped ZnO and S1:(0.1%), S2:(0.3%), S3:(0.5%); S4: (1%); S5: (3.0%); S6: (5.0%) and S7: (7.0 %) B doped ZnO respectively.” Please explain to me what percentage of doping this is (atomic or mass)? What were the volumes of the solutions mixed together?

  • All of the atomic doping percentages are calculated with XPS and results are shown in Table 3.

Reply: The results cannot be read from Table 3. Please present the data reliably in the table.

4. Crystallographic properties of the pure and B-doped ZnO films: Were the obtained ZnO rods monocrystalline ? Please compare the SEM results (Fig. 1) with the average crystallite size results obtained. Did the authors convert the actual dopant content against the formula for nominal dopant content? How can you explain the obtained difference in doping values (nominal vs. actual).

  • The obtained ZnO rods are polycrystalline.

The figure shows basically the actual dopings obtained from the XPS and the nominal values. The plot is included as inset in Figure 3 within the manuscipt.  The plot the nominal values and the actual dopings follow almost a linear variation. However, nominal and actual doping ratios are not expected to be exactly the same.

Reply: Please define for me an dopant of Zinc oxide expressed as a volume percentage.

5. Introduction: Please indicate/emphasise the novelty of your work more clearly. Is the novelty of the work the concentration of the dopant in the samples obtained?

Round 3

Reviewer 2 Report

The images of Fig. 4 were too small to identify each lines. I recommend the authors to make a spell check.   

Author Response

Thank you very much for the suggestions/corrections.

The Fig. 4 is revised and submitted with 600 Dpi resolution.

The text is checked and some spelling mistakes are corrected.  

Reviewer 3 Report

The dopant concentrations are inserted in the abstract. “The boron concentrations are 0.1, 0.3, 0.5, 1.0, 3.0, 5.0, and 7.0.”

Reply: Please explain to me how the author calculated the dopant value as a volume percentage. This is the first time I have come across a volume percentage of a dopant. General comment: % can be applied to dimensionless quantity.

  • The following relation is used while mixing two different solutions regarding volume

(v/v)% =  ( volume of dissolved solution(mL))/(Volume of solution(mL) ) x100

As it is seen from the relation it is dimensionless quantity multiplied by 100. Thus, we believe that “%” can be used. To make it more clear for the reader, we give more details in the experimental section in the revised manuscript.

Such calculation for “coumarin additive into ZnO” has already been published by Dr. Cetinkaya (one of the corresponding author of the present manuscript) in the Journal of Physics and Chemistry of Solids 74(2013) 565–569.

Response: The law of conservation of volume does not exist. There is a law of conservation of mass and constancy of composition. Please convert your results to atomic percent according to the formula:

B%=nB/(nZn+nB)*100%,

wher n=ms/M, n - number of moles, ms - mass of the substance, M - molar mass

Please also recalculate the actual dopant content in Table 3 according to the formula given.

Author Response

The calculation is done according to the Reviewer's suggestion and inserted in Table 3.